# A type III secretion system is required for *Bordetella atropi* invasion of host cells *in vivo*

**Tuan D. Tran, Serena J. Meadows-Graves, Amanda R. Haio, Alexander I. Varga, Robert J. Luallen** *

Department of Biology, San Diego State University, San Diego, California, United States of America

* rluallen@sdsu.edu

## Abstract

*Bordetella atropi* is an intracellular bacterial pathogen that infects the intestinal epithelia of the nematode host *Oscheius tipulae*. We previously showed that the bacteria use filamentation as a novel cell-to-cell spreading mechanism once inside the intestinal cell. However, how the bacteria invade the host cells and what factors contribute to *B. atropi* infection process remain unknown. In this study, we investigate the roles of type III (T3SS) and type VI secretion systems (T6SS) in *B. atropi* pathogenesis, which are employed by many bacterial pathogens, both extracellular and intracellular, to deliver effectors that manipulate host physiology to their advantage. We found that the two T6SSs encoded in *B. atropi* genome played no obvious roles in the invasion or intracellular spreading. In contrast, a T3SS was required for intestinal cell invasion. T3SS mutants showed loss of host cell protrusions from the apical surface that normally engulf invading wild type bacteria, as seen by both electron microscopy and confocal fluorescent microscopy. These protrusions bear morphological similarities to membrane ruffles triggered by the T3SS-mediated invasion seen in other pathogens such as *Salmonella* and *Shigella spp*. Additionally, we conducted dual transcriptomics and saw upregulation of T3SS *in vivo*, along with several putative effectors and the virulence regulator BvgS of the genus Bordetellae. We knocked out these effector candidates and found that deletion of one of these genes, *deiA* (decreased invasion protein A), leads to a reduction in the number of invasion events and overall percentage of infected animals in the population. In addition, deletion of the virulence regulator *bvgS* resulted in a complete loss of *B. atropi* invasion, suggesting it may regulate T3SS for host cell invasion.

## Author summary

The free-living nematodes, *C. elegans* and its related species, are useful models to study host-pathogen interactions in whole animal settings since they share many similarities in immune signaling pathways to mammals. We added

**Data availability statement:** The B. atropi strain LUAb4 complete genome sequencing project has been deposited at DDBJ/ENA/GenBank under the accession GCA_021325795.1, with BioProject PRJNA735517 and BioSample SAMN19587788. The dual RNAseq data seta has been deposited at DDBJ/ENA/GenBank under the accession Bioproject PRJNA1279217, with B. atropi infected JU1501 adults under BioSample SAMN49467810 and B. atropi infected JU1501 L1 larvae under BioSample SAMN49467811.

**Funding:** This work was funded by NIH grant R35 GM146836 and NSF IOS CAREER grant 2143718 to RJL. The funders had no role in study design, data collection and analysis, decision to publish, or preparation of the manuscript.

**Competing interests:** The authors have declared that no competing interests exist.

a new pathogen representative of a major class, intracellular bacteria, to the list of pathogens that can be studied using these animals. The new pathogen, *Bordetella atropi*, belongs to the *Bordetella* genus that includes important human pathogens like *B. pertussis*, the causative agent of whooping cough. We showed that *B. atropi* requires a specialized machinery known as type III secretion system to manipulate host intestinal epithelial cells to form distinctive morphological structures like those observed during invasion of *Salmonella* or *Shigella* species. These structures wrap around and engulf the invading bacteria, eventually lead to the uptake of the bacteria into host cell cytoplasm, and thus result in successful invasion of host cells by the pathogen. We also identified a gene that we named *deiA* for *de*creased *i*nvasion protein A whose protein product contributes to the capacity of the pathogen to invade host cells. Our results established a system for studying intracellular bacterial pathogenesis in whole animals, potentially with high conservation at the mechanistic level to mammalian systems.

## Introduction

*C. elegans* and related nematodes represent attractive model organisms to study infection processes and innate immunity, with many of their innate immune signaling pathways found to be largely conserved with those in mammals [1]. These organisms are amenable to a large repertoire of genetic tools and microscopy techniques that allow the observation of novel interactions between the pathogens and host cells in the context of a whole animal. For example, the fungal-like pathogen, *Nematocida parisii* was observed to cross the basolateral membrane of intestinal cells leading to host cell fusion [2]. Orsay virus infection was also observed to cause fusion of neighboring intestinal cells [3]. Another example is the recent observation that a host factor affects microsporidia spore orientation relative to the apical surface of host intestinal cells for efficient invasion of the sporoplasms [4].

A variety of natural pathogens that have been isolated from wild, free-living nematodes, including bacteria, eukaryotic parasites, and viruses [5]. Until recently, however, a major category of pathogens lacked representation in free-living nematodes, intracellular bacteria. Various clinically-relevant intracellular bacteria have been previously modeled in nematodes to study their pathogenesis, including *Salmonella enterica* Typhimurium [6], *Burkholderia* spp. [7,8], and *Listeria monocytogenes* [9]. While these models were utilized to identify factors required for nematode killing, including some known virulence master regulators and genes involved in bacterial metabolism and quorum sensing, these pathogens reside in lumenal space of the intestine and do not appear to carry out their intracellular lifestyle as observed in mammalian hosts. We recently discovered a facultative intracellular bacterial pathogen, *Bordetella atropi*, that infects the microscopic free-living nematode *Oscheius tipulae*. This bacterial pathogen has a novel infection paradigm. After intestinal cell invasion, *B. atropi* utilizes host nutrient sensing to trigger

replication without septation, resulting in filamentation. These bacterial filaments are then utilized to push through the lateral intestinal membranes for cell-to-cell spreading [10]. However, *B. atropi* filamentation was not required for the initial invasion of host cells, providing a unique opportunity to gain insights into intracellular bacterial invasion in a whole animal setting.

Bacterial pathogens are known to utilize different secretion systems to export a diverse array of exoproteins to support their growth and pathogenesis. Many species usually encode more than one type of secretion system, and each type may be present in more than one copy in the genome. The exoproteins, once secreted into extracellular environment or injected directly into host cells, can affect host cell physiology that ultimately allows the bacteria to adapt throughout different stages of infection or to different tissues and diverse hosts [11]. For extracellular bacterial pathogens, these exo-proteins may prevent phagocytosis by host immune cells [12,13], subvert host immune signaling [14], and mediate attach-ment to host cells and colonization [15]. For intracellular bacterial pathogens, these virulence proteins can also manipulate cellular processes to facilitate their entry into the intracellular environment and subsequently, the formation of a replicative niche. The direct injection of these virulence factors is achieved mainly by specialized machineries known as type III- and type VI secretion systems (T3SS and T6SS, respectively) [16–18]. The T3SS of many human pathogens, including *Sal-monella* [19] and *Pseudomonas aeruginosa* [20], were shown to be required for their virulence. For instance, *Salmonella* Typhimurium has two T3SSs located in two distinct *Salmonella* pathogenic islands (SPIs). The SPI-1 T3SS is used for invasion of epithelial cells [21,22] while the SPI-2 T3SS is used for intracellular vacuolar replication and systemic infection [23]. *Burkholderia pseudomallei* harbors three T3SSs and six T6SSs, with T3SS-3 being implicated in nonphagocytic cell invasion and vacuolar escape [24,25] and T6SS-5 being required for host cell fusion and the formation of multinucleated giant cells [26].

By contrast, a role for T3SS-mediated pathogenesis in nematodes has been less well-established and do not appear to reflect the situations seen in mammalian cell culture. For instance, the T3SS has been implicated in *P. aeruginosa* extra-cellular infection in humans [27], but appears dispensable for *C. elegans* killing [28,29]. Similarly, enteropathogenic *Esche-richia coli* (EPEC) and enterohemorrhagic *E. coli* (EHEC) require a T3SS for their attaching and effacing phenotypes in mammalian hosts, but lumenal colonization and host microvillar effacement in *C. elegans* appear to be independent of the T3SS machinery [30,31]. In some cases, a role of T3SS has been suggested for pathogenesis in *C. elegans*, such as T3SS-triggered degradation of the host intestinal transcription factor ELT-2 by *B. pseudomallei* [32], or increased survival of *glp-4* animals when infected with *S. enterica* Typhimurium T3SS mutants [33]. However, these facultative intracellular bacteria remain as extracellular pathogens in *C. elegans*, suggesting the use of the T3SS in these scenarios may repre-sent a different mode of infection when compared to their intracellular lifestyle in mammals. Much less is known about the roles of T6SS in bacterial infection in nematodes, except for studies reporting improved survival of the animals fed with T6SS mutants compared to wildtype bacteria [34,35].

In this study, we investigated the roles of the T3SS and T6SS in *B. atropi* infection in the nematode *O. tipulae*, with deletion of the T3SS showing a discernable virulence phenotype. We found that the T3SS is required for *B. atropi* invasion into intestinal epithelial cells, but apparently dispensable for the microvilli effacement. *B. atropi* T3SS-mediated invasion induces host processes that engulf individual bacteria and have an appearance reminis-cent of host membrane ruffles induced by mammalian intracellular bacteria such as *Salmonella* and *Shigella*. Fol-lowing invasion, the bacteria escape into the cytoplasm and remain as cytosolic bacteria prior to filamentation. We utilized dual transcriptomics to identify candidate genes used for the invasion process and found that an upstream virulence regulator, *bvgS*, and a putative proline-rich effector gene *deiA* (decreased invasion protein A) are also required for normal host cell invasion. Our results demonstrate that the T3SS mediates *B. atropi* invasion of nem-atode intestinal epithelial cells and this process displays significant morphological similarity to that of mammalian intracellular bacterial pathogens, suggesting potential high level of mechanistic conservation between distant bacte-rial pathogen-host systems.

## Results

### 1.  Identification of secretion systems in *B. atropi*

To dissect the contribution of different secretion systems to *B. atropi* infection, we first performed genomic analysis to identify putative loci encoding for secretion apparatuses. Using a pipeline of available software and databases (see Methods), we identified core components for several secretion systems, including one type I (T1SS), one type II (T2SS), one type III (T3SS), and two type VI (T6SS-1 and T6SS-2) (Fig 1A). We also found multiple autotransporter proteins potentially related to the type V secretion system and genes encoding components of other more general secretion pathways SecYEG and TatABC located throughout the genome.

Mammalian intracellular bacterial pathogens employ predominantly the T3SS and, to a lesser extent, T6SS for an intracellular lifestyle [16,36]. Thus, we focused on studying the roles of the single T3SS and the two T6SSs in *B. atropi* infection. First, we knocked out the T6SSs by truncating the regions encoding for the T6SS sheath and needle subunits – *tssBC-1* and *tssBC-2* – either individually to create ΔT6SS-1 and ΔT6SS-2 strains, respectively, or in combination to generate the double knockout ΔT6SS-1/2. For wild type *B. atropi* infection, infected animals typically exhibit short bacterial filaments or small clusters of bacteria at 16 hours post inoculation (hpi), long filaments at around 24 hpi corresponding with intracellular spreading, and eventually a reversion back to a multitude of coccobacilli by 48 hpi [10]. We thus chose to examine bacterial phenotypes at 24 hpi for the infection with these mutants as this time point can inform us about the invasion, filamentation, as well as spreading capacity of the mutants. For simplicity, we refer to infected animals as those showing intracellular invasion, whereas uninfected animals show no intracellular invasion, but may have colonization in the lumen. At 24 hpi, both the individual knockouts ΔT6SS-1 and ΔT6SS-2, as well as the double knockout ΔT6SS-1/2 could infect and exhibit a filamentation phenotype similar to the wildtype at 24 hpi (Fig 1B). Furthermore, there was no significant difference in the percent of the host population showing infection in the T6SS mutants compared to the wildtype, in or around ~37% (Fig 1C), suggesting that these knockouts did not affect the gross infectivity nor filamentation capacity of *B. atropi*.

After intestinal cell entry, *B. atropi* undergoes a morphological change from coccobacilli into long filaments of up to ~130 μm in length as a mechanism for cell-to-cell spreading [10]. Several studies have reported the roles of T6SS in compromising the host cell membranes by *V. cholera* (VasX) and *B. pseudomallei* (VgrG5) [26,37], a potential crucial step for intracellular spreading in host tissue. Given the T6SS-2 locus contains both genes encoding VasX motif containing protein as well as several VgrG homologs (*tssI*-2, -3, and -4) (Fig 1A), we aimed to test if T6SS affects the bacteria infectious capacity by altering the capacity of filaments to spread between cells. To this end, we measured the normalized anterior-posterior infection length of the ΔT6SS-1/2 [10]. We did not observe a significant change in normalized anterior-posterior infection length in these mutants compared to WT bacteria (Fig 1D), suggesting that T6SS did not affect the spreading capacity of the bacteria. Overall, it appears that T6SS is not involved *B. atropi* invasion or the intracellular spreading of filaments during infection in the nematode host *O. tipulae*.

### 2.  *B. atropi* invasion requires a functional type III secretion system

We next sought to determine the roles of T3SS during *B. atropi* infection. *B. atropi* contains all core components of the T3SS machinery with a single class Ib effector chaperone (*cesT*) responsible for secretion of multiple T3SS effectors [16,38] (Figs 1A and 2A). To knock out the T3SS, we targeted *sctC* encoding a subunit for the intermembrane complex of T3SS machinery. This knockout resulted in zero animals showing infection, despite having bacteria in the intestinal lumen throughout the duration of the experiments, suggesting that *B. atropi* requires a functional T3SS for invasion (Fig 2B). To further test this hypothesis, we knocked out other genes in the T3SS operon, including *sctB* and *sctE* that have structural homology to translocon pore subunits (Fig 2A). Knockouts of *sctB* and *sctE* phenocopied the noninvasive phenotype of Δ*sctC* at 48 hpi, a time point where *B. atropi* infection is normally at the late stage. In all cases, the mutant bacteria resided and colonized the lumen but could not infect host intracellularly (Fig 2B).

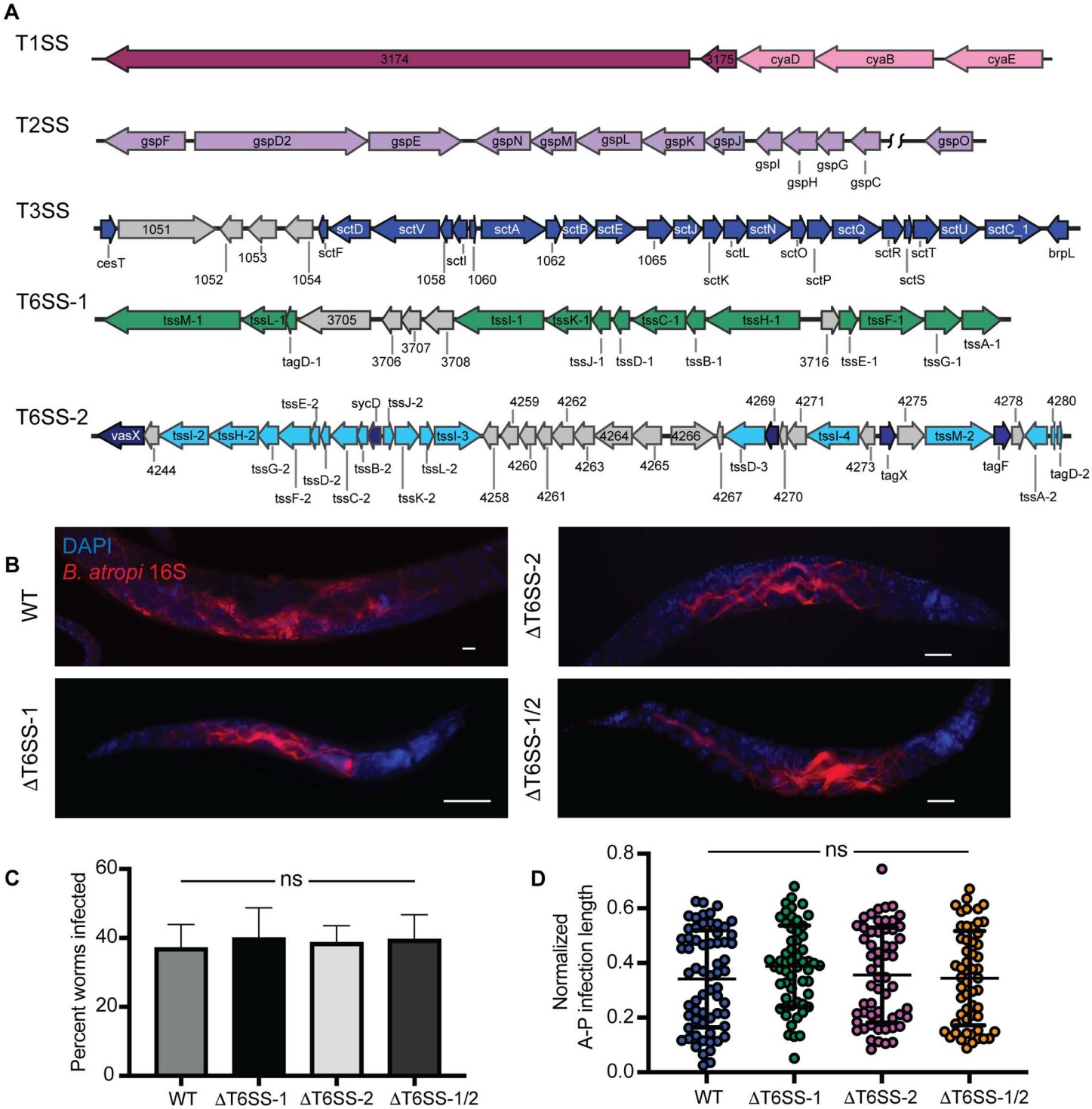

**Fig 1. The T6SS of *B. atropi* is not required for establishing infection or intracellular spreading. (A)** Schematic of secretion system loci with annotated component genes. **(B)** Representative images of worms infected with knockout strains of T6SS. **(C)** Quantification of percentage of worms infected with each strain (n = 200 worms over 2 independent experiments). **(D)** Normalized anterior-posterior infected length of *B. atropi* WT and T6SS knockout strains *in vivo* (n = 60 worms per strain from 2 independent replicates). Scale bars are 10 μm. Graphs show means with SD, ns, nonsignificant by Mann-Whitney U test.

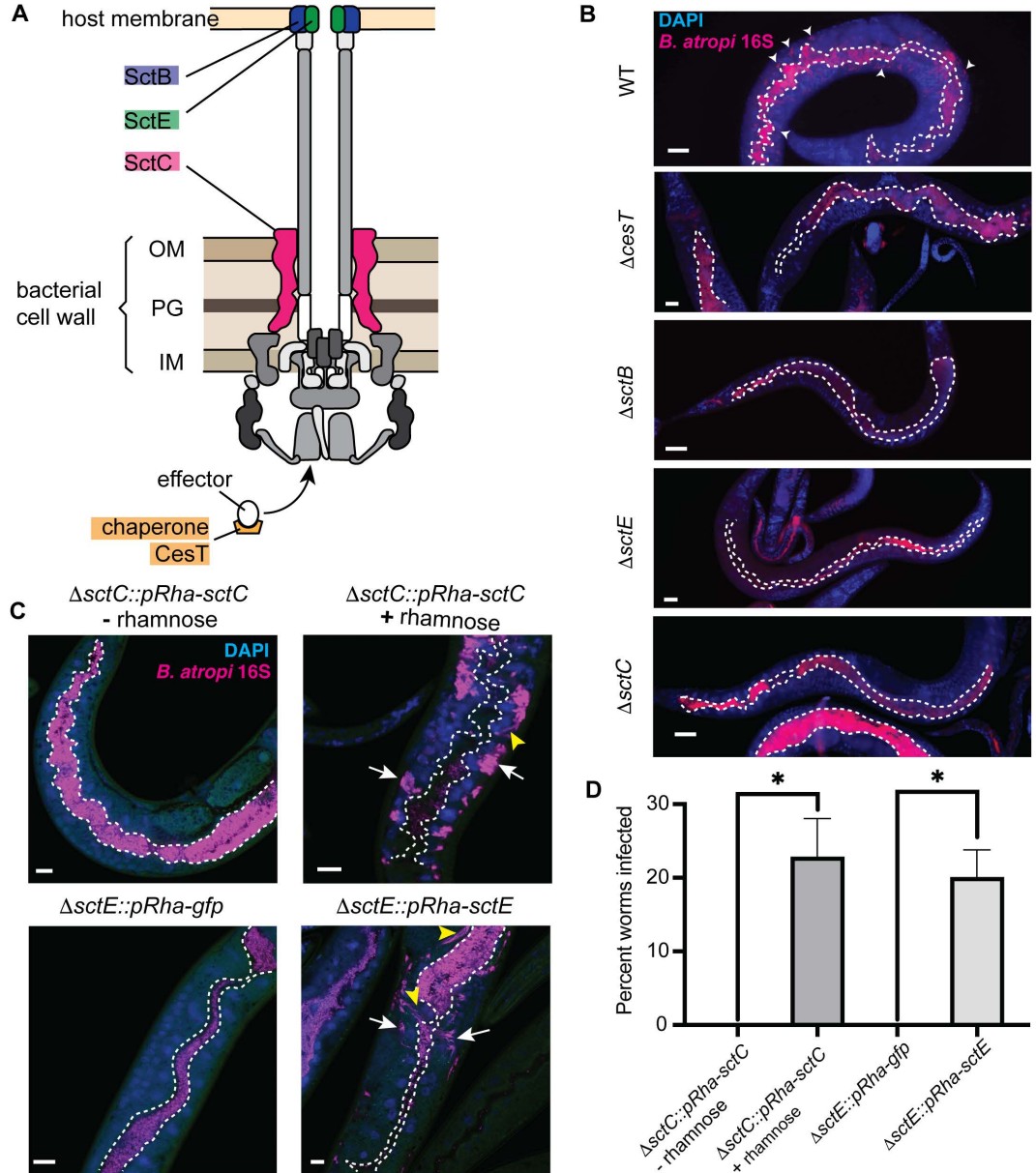

**Fig 2. Type III secretion system is required for *B. atropi* invasion of host epithelial cells. (A)** Schematic of *B. atropi* T3SS machinery. OM, outer-membrane, IM, inner membrane, PG, peptidoglycan. Highlighted in colors are subunits used for knockout strains. **(B)** Representative images of worms infected with Δ*cesT*, Δ*sctC*, Δ*sctB*, Δ*sctE* compared to WT. White dotted lines delineate the intestinal lumen. Arrowheads in WT indicate infection foci. **(C)** Representative images of animals infected with complemented strains of *sctC* and *sctE* compared to corresponding controls at 24 hpi. Dotted lines outline the lumenal space, white arrows indicate cluster of intracellular bacteria, yellow arrowheads indicate bacterial filaments. **(D)** Percentage of worms invaded in the complemented strains when induced on NGM plate containing 3 mg/mL L-rhamnose (n = 30 per condition over 2 independent replicates). Scale bars are 20 μm in B and 10 μm in **C**. Graphs show means with SD, p = 0.0253 for *sctC* and 0.017 for *sctE* by unpaired two-tailed t test.

Interestingly, it has been reported that in *Shigella*, the T3SS apparatus alone without any effectors can mediate sufficient vacuole lysis and phagosomal escape, raising the possibility that T3SS machinery *per se* may have additional, species-dependent "effector-like" function other than just a device to deliver effectors [39]. Thus, to test whether *B. atropi*

invasion requires the translocation of effectors, we knocked out the single effector chaperone-encoding gene *cesT* and infected animals with this strain. Similarly to T3SS apparatus knockouts, we did not observe any animals with intracellular bacteria, suggesting that *B. atropi* invasion does require effectors and not just the T3SS apparatus (Fig 2B, Δ*cesT* panel).

To confirm the role of the T3SS in invasion, we conducted a complementation assay by expressing *in trans* in the Δ*sctC* background the WT copy of *sctC* under regulation of a rhamnose-inducible plasmid and examine the infection at 24 hpi. In the absence of the inducer, all of the animals showed bacteria confined within the lumenal space of the intestine, while addition of the inducer restored the infection to an average rate of 22.3% with multiple clusters of bacteria being detected close to the basal side of host intestinal cells (Fig 2C top panels and Fig 2D). Similarly, we complemented Δ*sctE* with either WT copy of *sctE* or *gfp* in the same inducible system and found similar result in which the WT copy of *sctE* could rescue the infection whereas the *gfp* control could not (Fig 2C bottom panels and Fig 2D). Additionally, we also observed instances of bacterial filamentation in the rescued strains during infection (Fig 2C, yellow arrowheads), suggesting that the bacteria exhibit WT infectious characteristics. Altogether, these data support T3SS requirement for *B. atropi* invasion that is facilitated by translocation of T3SS effectors into host cells.

## 3. Transmission electron microscopy reveals microvilli effacement and host cell protrusions that engulf bacteria in the lumen

Based on the results of the genetic knockouts of *B. atropi* T3SS, we sought to better visualize the invasion process of *B. atropi* at a higher resolution, utilizing a timepoint when could putatively capture internalization events by bacteria. To this end, we conducted transmission electron microscopy (TEM) on animals infected with either WT bacteria or the T3SS mutant strain Δ*sctC*. We performed pulsed infection of animals with each pathogen strain for two hours and harvested infected animals at 22 hours post-infection (hpi). We chose this time point based on our previous study in which a pulse infection will result in the earliest detection of invasion at 16 hpi [10]. Since TEM was carried out on bulk nematode pellets, we reasoned that 22 hpi would potentially allow us to observe invasion events at a higher probability.

From TEM images, we observed that animals infected with either WT or T3SS mutant Δ*sctC* strain appeared to have shorter, distorted, and irregular distribution of microvilli compared to uninfected animals grown on normal food source OP50–1. To quantitatively determine if *B. atropi* causes disruption of the brush border of host intestine, we measured the length of microvilli in TEM images that captured unique cross sections of the intestine in which the microvilli appear perpendicular to the apical surface. From 45 unique sections of different angles, we identified five images of uninfected control samples, eight images of animals infected with WT bacteria, and five images of animals infected with Δ*sctC* strain that satisfied this criterion. We defined the microvillus length as the distance from the apical membrane to the tip of the microvillus, measuring only microvilli that appear connected to the plasma membrane (i.e., forming continuity to the intestinal cell cytoplasm); microvillar fragments are likely from a different plane and do not reflect the true lengths (Fig 3A). Using these criteria for measuring microvillar lengths, we found that WT bacteria cause a moderate but significant reduction of average microvillar length compared to uninfected samples (Fig 3B), from ~0.55 μm in uninfected samples to ~0.35 μm in *B. atropi* infected samples. Surprisingly, our T3SS mutant Δ*sctC* resulted in even further shortening microvilli (to ~0.3 μm), indicating that the effacement of the microvilli may be independent of a functional T3SS in the host *O. tipulae*.

We analyzed the TEM images of WT *B. atropi* infection to potentially see early invasion events. We found 4 events where a single bacterium appeared to be being engulfed by a host cell process, which protruded into the lumenal space relative to the general apical surface level of neighboring cells (Fig 3C). Interestingly, certain portions of these incoming bacterial cells were already in contact with host cell cytoplasm while the host cell protrusions appear to enclose them to form the uptake vesicles. This observation hints at a scenario where the bacteria perhaps could trigger engulfment and instantaneously escape from the forming vesicles. This quick escape into the cytoplasm is congruent with the bacterial phenotype classification from TEM. Specifically, when we categorized the bacteria into different groups based on locations

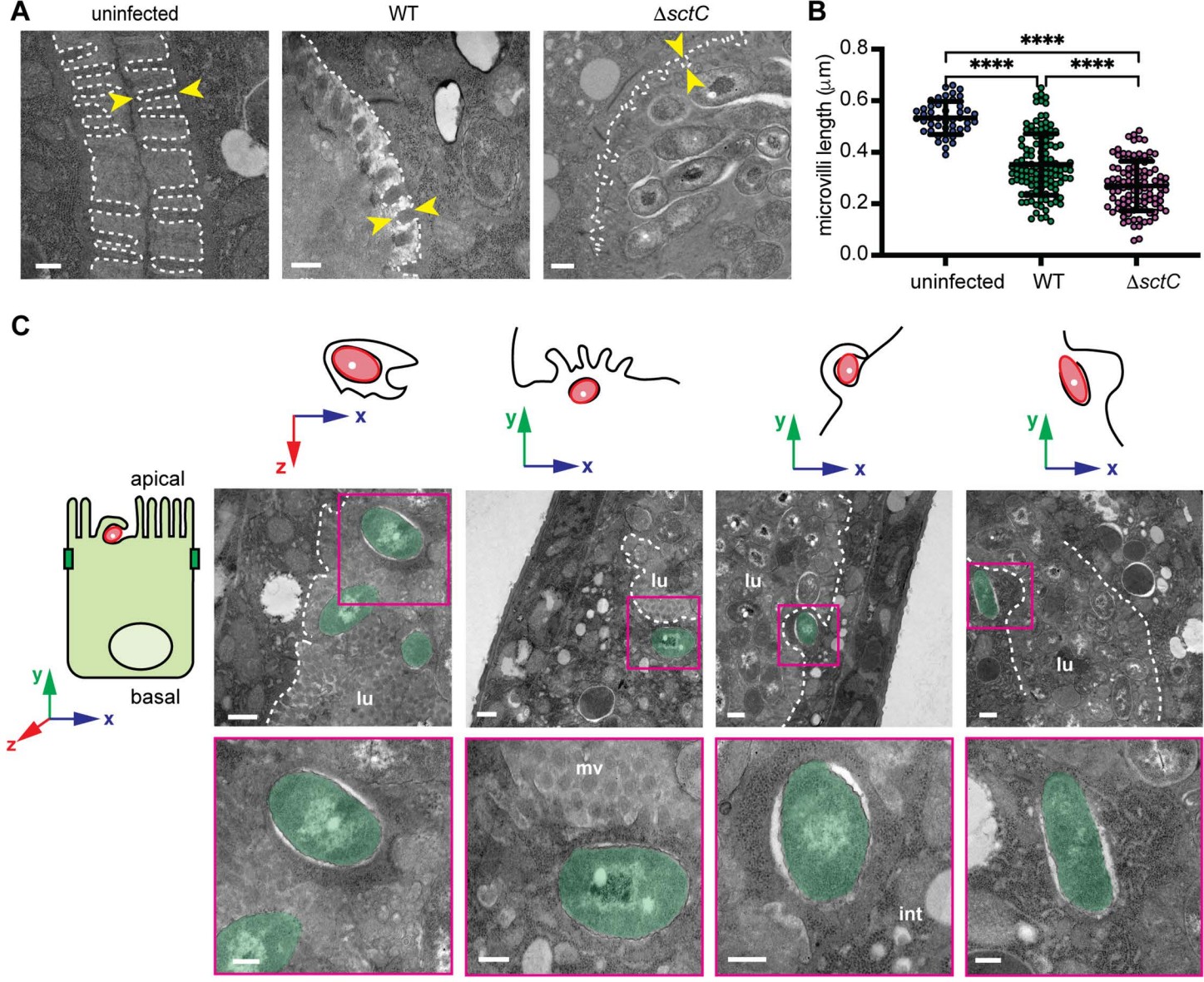

**Fig 3. *B. atropi* infection induces effacement of the microvilli and host cell protrusions. (A)** Representative TEM images of animals uninfected or infected with either the WT or T3SS mutant. Microvilli in the same plane as the apical membrane, but not microvilli fragments disconnected from cytoplasm, are outlined and measured (white dashed line). The length of a microvillus is defined by the yellow arrowheads. **(B)** Microvillus length of samples from A. Each dot represents the measurement of a single microvillus. Scale bars are 250 nm. p < 0.0001 (****) by Mann-Whitney two-tailed U test. **(C)** TEM images of WT bacteria at 22 hpi showing bacteria being engulfed by intestinal cell extensions. The axes above each image shows the orientation of the TEM image to the reference axis of the intestinal cell (on the left). For example, the left TEM image is a top-down veiw of bacterial engulfment, while the remaining images are side views. lu, lumenal, mv, microvilli, int, intestine. Scale bars are 250 nm in A and in lower panel of insets in C, and 500 nm upper panel in C.

and phenotypic appearances from 45 unique sections of animals infected by WT bacteria, among the 36.4% of intracellular bacteria, 76.9% were in direct contact with host cytoplasm, and 23.1% of them were surrounded in a white halo with no clear membrane around them (intracellular halo) (S1 Fig), indicative of bacteria not residing in membranous compartments formed from the engulfment.

## 4. The T3SS is required for *B. atropi* to induce host cell protrusions

To further validate the host cell protrusion phenotype from our TEM data, we employed an alternative approach to examine bacterial invasion in a sufficiently large population of infected animals. We utilized the dye carboxyfluorescein diacetate tetrazine (CFDA-Tetrazine) to stain the intestinal cell cytoplasm to visualize any host cell protrusion. We observed multiple protrusions extending to the lumenal space and wrapping around individual bacteria, in agreement with TEM observations (white arrowheads in Fig 4A, and yellow arrowheads Fig 4B orthogonal projections). In contrast, these protrusions are absent in all animals colonized with Δ*sctC* control under the same conditions, as the intestinal apical surface of animals infected with Δ*sctC* strain exhibits smooth and neat delineation from the lumen (0/20 for Δ*sctC* as compared to 19/20 for WT in two independent replicates) (Fig 4B - 4C), suggesting that these cellular extensions are likely an effect of *B. atropi* T3SS function. Altogether, our TEM and intestinal staining data suggest that *B. atropi* employs T3SS to trigger host cells to form protrusions that ultimately lead to bacterial internalization.

We also attempted to assess whether invading bacteria make direct contact with host cytoplasm immediately after engulfment using this same experimental paradigm. CFDA-Tetrazine in the host cytoplasm would react specifically with TCO-pretreated bacteria through fast click-chemistry [40] if the invading bacteria are in direct contact with host cytoplasm as soon as they trigger host cell engulfment. However, CFDA-Tetrazine pre-stained animals infected with either TCO-pretreated WT bacterial strain or TCO-pretreated Δ*sctC* strain resulted in no observed signal colocalization (Fig 4B), potentially due to low CFDA-Tetrazine signal on bacterial surface.

## 5. *B. atropi* upregulates expression of T3SS and putative effectors *in vivo*

Since *B. atropi* invasion requires the translocation of T3SS effectors, we reasoned that such effectors could be upregulated during infection. To investigate this possibility, we conducted dual transcriptomics on bacteria from infected animals at 28 hpi (*in vivo*) and bacteria grown in LB (*in vitro*). Out of 5582 protein coding genes in *B. atropi* genome, we found 698 genes significantly upregulated and 260 genes significantly downregulated *in vivo* compared to *in vitro* at a false discovery rate (FDR) of 0.1% (S1 and S2 Tables). We performed principal component analysis (PCA) on differentially expressed genes (DEGs) to evaluate the variance of bacteria transcriptome under in vivo and in vitro conditions (Fig 5A). Ninety eight percent of variance in samples were represented on both PC1 and PC2, with 97% of variance explained by PC1, indicating major shift in *B. atropi* gene expression upon exposure to host.

Among the in vivo upregulated genes, those encoding for T3SS apparatus show the highest fold changes compared to in vitro conditions, in agreement with the requirement of T3SS and translocation of effectors for invasion (Fig 5B). These include translocon components *sctE*, *sctA*, *sctB*, the needle subunit *sctF*, the membrane complex subinits *sctC* and *sctD*, as well as the effector chaperone *cesT*. In addition, we also detected upregulation of genes encoding the virulence master regulator *bvgAS* of the Bordetellae genus and the putative *brpL* σ factor, all of which have been shown to be required for expression of T3SS in *B. pertussis* and *B. bronchiseptica* [41]. This observation hints that *B. atropi* may sense the host environment through the two component system BvgAS that in turn regulates the expression of T3SS machinery and effectors for invasion and subsequent stages of infection (i.e., establishment of replicative niche and intracellular spreading after filamentation).

To idendify potential T3SS effectors employed by *B. atropi* during infection, we first scanned the genome for known virulent effector motifs from the KEGG database [42], leading to the detection of putative T3SS effector homologs, including ExoY [43], YopJ [44], SseL [45], Avrpphf-orf2 [46], and NEL domain-containing proteins [47]. Since invasion by intracellular bacterial pathogens often involves manipulating host cell cytoskeleton through actin dynamics, often by proline-rich motif (PRM) containing proteins [48], we sought to determine if any of PRM-containing protein coding genes are also highly upregulated during *B. atropi* infection that could potentially act as T3SS effectors. To this end, the remaining hypothetical proteins after KEGG motif search were then subjected to compositional bias analysis for proline using fLPS2 [49]. The combined list of proteins from the motif search and proline compositional bias analysis were compared against in vivo

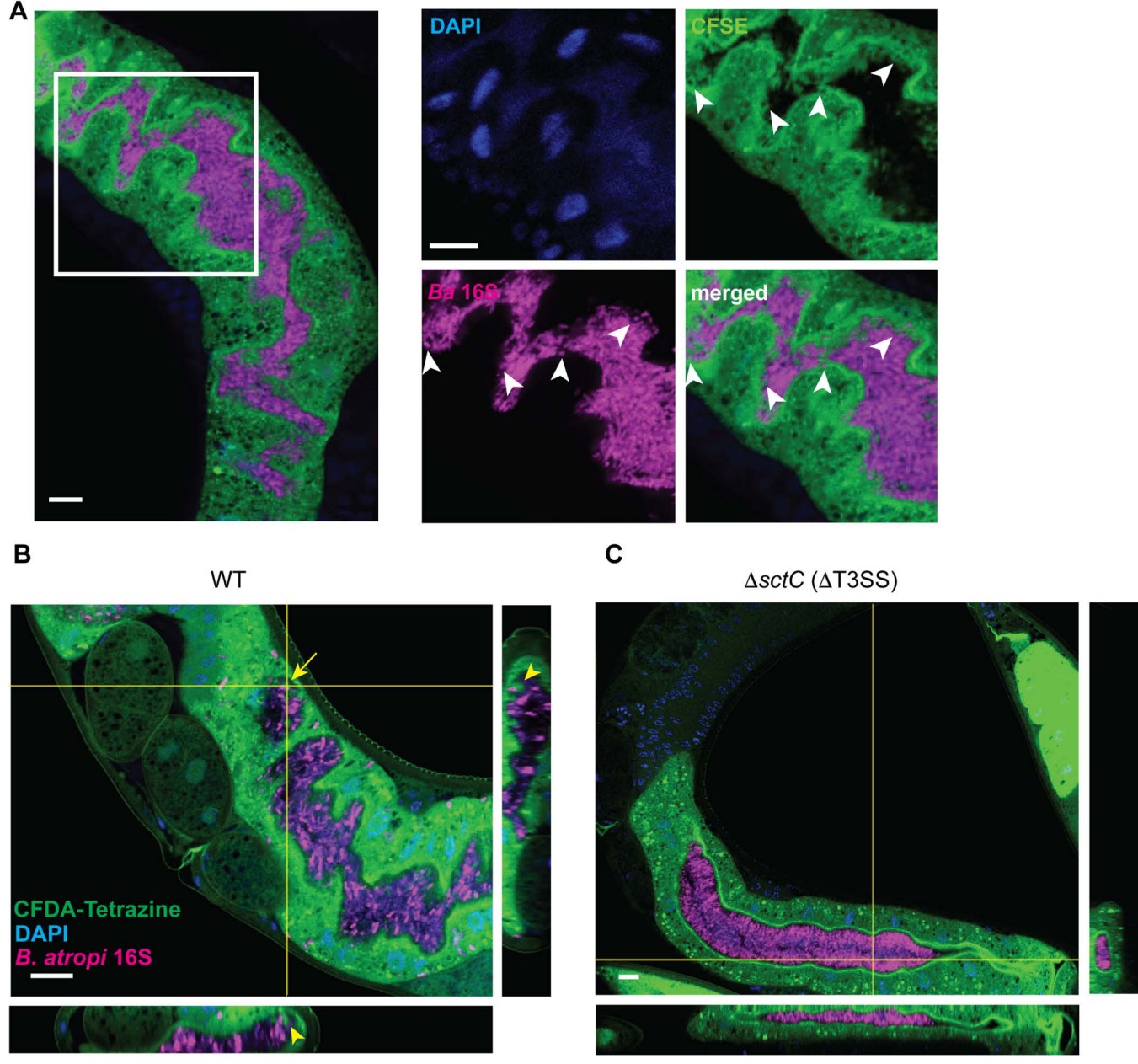

**Fig 4. *B. atropi* invasion induces host cell membrane protrusions in a T3SS dependent manner. (A)** (Left panel) Representative image of cytoplasmic material extension can be seen in the lumen of an infected animal. (Right panels) Zoomed in of inset in A showing protrusions containing cytoplasmic materials extending from apical surface (white arrowheads in CFSE channel) and wrapping around bacteria (white arrowheads in *Ba* 16S channel). (**B-C**) Representative confocal microscopy image of a worm infected with WT bacteria with projections showing host cell protrusions from apical surface extending to lumen to engulf a single invading bacterium B and a worm infected with T3SS mutant exhibiting "smooth" intestinal delineations and absence of host cell protrusions. Scale bars are 10 μm.

upregulated DEGs (S1 and S2 Tables) with more than 4 fold changes and FDR<0.1% to identify genes that are highly upregulated in vivo and either a known T3SS effector homologs or proline-rich domain containing proteins (Fig 5C). This resulted in a list of 54 PRM-containing candidate T3SS effectors (S3 Table), from which we chose three candidate genes

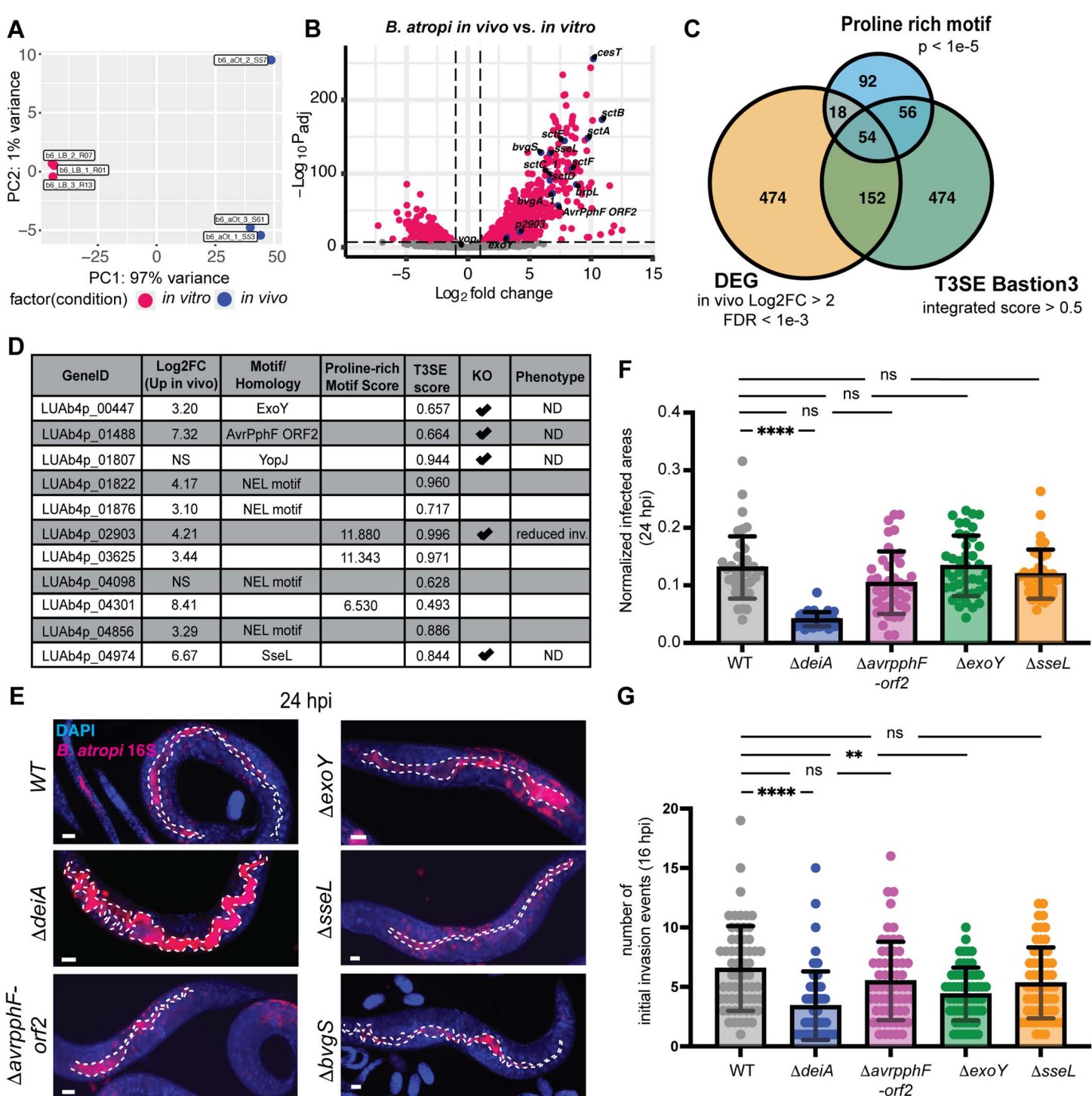

**Fig 5. B. atropi upregulates T3SS and putative effectors in vivo. (A)** Principal component analysis of *B. atropi* in infected animals compared to in vitro. **(B)** Volcano plot of DEGs with log2 fold change >2 and FDR<0.1% (in red). Genes encoding T3SS apparatus components, virulence regulators, and effector candidates are indicated in blue. **(C)** Venn diagram of DEGs in B, hypothetical proteins with proline compositional bias by fLPS2, and hypothetical proteins predicted to be T3SS effectors by Bastion3. **(D)** Combined list of putative T3SS effectors. **(E)** Representative images of knockouts of select putative effectors from D. **(F-G)** Normalized infected areas **(F)** and number of initial invasion events **(G)** in animals infected with putative effectors knockouts in E. Graphs show means with SD over 2 independent replicates, ****, p<0.0001, **, p=0.0063, ns, non-significant by Kruskal-Wallis test followed by Dunn's tests. Scale bars are 10 µm.

with the highest proline-rich scores to form a combined list of putative T3SS effectors with other homologs of known T3SS effectors (Fig 5D).

We selected several genes from our T3SS effector candidate list (Fig 5D) together with *bvgS* – the histidine sensor of the virulence master regulator BvgAS – to knock out and investigate their potential function in *B. atropi* infection by different metrics, including the number of initial invasion events at 16 hpi, the percentage of infected animals and normalized infected areas at 24 hpi. We found that knocking out *sseL* and *avrpphF-orf2* indvidually did not result in significant change in percentage of animals infected with *B. atropi* or their infectious capacity as measured by the normalized infected area (Figs 5E and S2A). By contrast, knockout of *bvgS* resulted in no infection (Fig 5E), similar to T3SS mutants, suggesting that the two-component system BvgAS may also act upstream and regulate the expression of T3SS operon like the classical *Bordetellae* [50]. Interestingly, we found that knockout of *exoY*, while not resulting in significant changes in the percentage of infected animals nor normalized infected area, led to a reduced number of initial number of invasion events. Interestingly, knockout of the PRM-containing protein encoding gene *LUAb4p_02903*, which we have named *deiA* (decreased invasion protein A) led to a significant reduction in invasion events at 16 hpi and infected area at 24 hpi (Figs 5F-5G and S2A). These effects observed in Δ*exoY* and Δ*deiA* are not due to a growth defect of these strains (S2B Fig), suggesting that these genes encode *bona fide* T3SS effectors. Complementation of Δ*deiA* strain with a WT copy, however, caused a severe growth defect (S3 Fig), suggesting that uncoupling *deiA* from its original regulation within the T3SS through (inducible) overexpression system is toxic to the bacteria. As such, we could not fully validate *deiA* as a T3SS effector through complementation. Future studies to the transcriptional regulation of T3SS during infection may provide additional information for demonstrating a direct causal relationship between *deiA* and the observed phenotype. Overall, our data support the requirement of a functional T3SS for invasion and intracellular infection of *B. atropi in vivo*.

## Discussion

In the current study, we elucidated the roles of *B. atropi* T3SS and T6SS during infection in its host *O. tipulae*. We found that the two loci encoding T6SSs did not contribute to a discernable outcome of *B. atropi* infection *in vivo*, whereas the T3SS was completely required for invasion and therefore, contributed at least to the first step of the intracellular lifestyle of the pathogen.

We found that T3SS is required for *B. atropi* invasion of host intestinal epithelia, likely by triggering host cells to form protrusions from the apical surface to engulf the invading bacteria. These protrusions are morphologically reminiscent of membrane ruffles triggered by T3SS during invasion by mammalian intracellular bacteria such as *Salmonella* or *Shigella spp.* in cell culture system [51–54], indicating potential similar host targets for induction of cytoskeletal rearrangement. Furthermore, our TEM data suggest that *B. atropi* may cause effacement of host intestinal microvilli, but the effacement is independent of T3SS. This appears to be different from mammalian models where microvillar effacement is a consequence of T3SS function during invasion [55,56], but in agreement with previous findings in *C. elegans* models of EHEC/EPEC [30,31] and *S. aureus* – a Gram-positive pathogen that does not encode a T3SS [57], suggesting that there may be multiple mechanisms that cause microvillar degradation in nematode-microbe interactions.

Our in vivo transcriptomics data showed upregulation of multiple genes encoding for T3SS apparatus components, as well as the master virulence regulators *bvgAS*. It has been shown that BvgAS controls the expression of T3SS operon in response to different environmental stimuli in the classical *Bordetellae*, including exposure to blood serum [58], $CO_2$ concentrations [59], glutamate starvation [60], and redox state [61]. It is therefore tempting to speculate that perhaps the lumenal environment of the nematodes with low pH (in the range 4–5) [62] and potentially a more oxidative environment [63] may be sensed by the two component system BvgAS, which in turn triggers the expression of T3SS operon and its effectors to coordinate invasion into host cells.

We also detected upregulation of several T3SS effector homologs; the individual knockouts of some of these putative effectors did not show significant alteration in *B. atropi* overall capacity to infect and spread in the host. This is likely due

to the complexity of network interactions and/or overlapping functions among the complete set of T3SS effectors under a particular condition (i.e., specific tissue or host), as has been suggested recently in case of *C. rodentium* [64]. We did, however, observe a slight reduction in the number of initial invasion events for Δ*exoY* despite having no significant changes in other infectivity aspects (i.e., normalized infected area and percent animal infected). ExoY is a nucleotidyl cyclase that has been implicated in interacting and modulating actin cytoskeleton [65–67], an effect that could potentially explain the reduced number of initial invasion events observed in the knockout.

Interestingly, one of our predicted T3SS effector candidates, *deiA*, displayed reduced infection phenotype upon deletion in all metrics we evaluated, including number of initial invasion events, normalized infected areas, and percent animals infected. Given the role of many PRM containing proteins in manipulating host cell cytoskeleton through actin dynamics [48], it is intriguing to consider that *deiA* may play a role in manipulating host physiology (i.e., triggering host cell engulfment), adhering to host epithelia, or establishing a replicative niche in the host cell. The exact mechanism of how this PRM-containing protein functions to facilitate the normal infection process of *B. atropi* warrants additional studies in the future. One interesting possibility is that *deiA* may facilitate the nucleation of actin to form filaments and act in concert with *exoY*, which could bundle actin filaments to form engulfment structure, and that could explain the smaller reduction in invasion for Δ*exoY* compared to Δ*deiA*. Additionally, we have identified greater than 50 additional T3SS PRM candidates (S3 Table) that could also play a role in *B. atropi* infection. Overall, our study demonstrates a system with a conserved mechanism of T3SS-mediate intracellular invasion to investigate common principles as well as potentially new host-pathogen interactions *in vivo*.

## Materials and methods

### Animal husbandry and synchronization

Wild-type *O. tipulae* strain JU1501 were grown on nematode growth media (NGM) plates with *E. coli* OP50–1 as normal food source. For experiments that need to use either first larval state (L1) animals or young adults, nematodes were synchronized as following: animals were grown until near starvation at which time the plates contain mainly gravid adults and L1 larvae, then collected in M9T (M9 media, 0.05% TritonX-100) in 15 mL conical tubes. Animals were allowed to segregate by gravity, with adult animals settling at the bottom of the tube after ~ 5 minutes. The supernatant contains mostly L1 animals, which then can be used for experiment or plated onto new NGM plates and allowed to grow for 3–4 days in incubator at 23°C to reach young adult stage for infection.

### Bacterial strain construction

In-frame deletion was carried out as described previously [10]. Briefly, about 1.5-kb fragments upstream and downstream of genes of interest (including the first and last 5–10 codons, respectively) were cloned and Gibson assembled together with the triple-digested vector pCVD443 (a gift from Dr. Nicholas Shikuma) using NEBuilder HiFi DNA Assembly Cloning Kit (New England BioLabs Inc., E5520S). Assembly mixture was then used to transformed *E. coli* SM10 lpir by electroporation (MicroPulser Electroporator, Bio-Rad). The parental *E. coli* SM10 strain was used to mate with wild-type *B. atropi*. Exconjugants were selected for first recombination event with antibiotics and subsequently second recombination event with 10% sucrose. Putative knockout colonies were confirmed for deletion of the gene of interest by colony PCR with primers flanking the gene.

Complemented strains were generated by mating the mutant strain with *E. coli* SM10 that had been transformed with plasmid pRha containing a rhamnose-inducible cassette that regulates expression of gene of interest (pRha was a gift kindly provided by Dr. Anca Segall).

### In silico identification of secretion systems

The *B. atropi* genome (GCA_021325795.2) was submitted to MacSyFinder-based TXSScan [68] and GhostKOALA [69] webservers to identify putative secretion systems. From this search, for any incomplete secretion system, we performed

BLAST search using genes encoding for these missing components from other species, followed by an AlphaFold prediction [70,71] and structural homology search with DALI [72] or inspection of superimposition of predicted structures and known structures from Protein Data Bank (http://www.rcsb.org/) [73] using PyMol software.

### *B. atropi* infection

For standard infections, bacteria were grown in an LB culture with appropriate antibiotics overnight. One milliliter of this overnight culture was spread on an NGM plate to cover the whole surface to make infection plates. Animals grown on normal food source OP50–1 were then plated on infection plates and kept in incubator at 23°C until harvest for downstream assays.

For pulse infections, after 2 hours on the infection plates, animals were harvested and washed 4 times, each with 1 mL of MTEG (M9 media, 0.05% v/v TritonX-100, 0.25 mM EDTA, 20 µg/mL gentamicin) for 20 minutes, quickly rinsed with M9T and replated onto OP50–1 seeded NGM plates.

For overlay infections, 1 mL of bacterial culture at $OD_{600}$ of 0.8-1.0 was spun down, washed 3 times, each with 1 mL of M9, resuspended in 1 mL of fresh M9 and overlaid on top of worms grown on OP50–1 seeded NGM plate. Plates were dried briefly and kept at 23°C.

Bacterial strains harboring the rhamnose-inducible promoter controlling the expression of gene of interest were grown in LB with 20 µg/mL gentamicin overnight and 1 mL of the culture was spread on an NGM-Gent plate (NGM, 20 µg/mL gentamicin) for 1 day before the assay. *O. tipulae* JU1501 adult animals were plated on these plates overnight to allow bacteria colonize the lumen of the nematodes, after which they were collected in M9T, quickly washed a few times in M9T until the liquid was visibly clear, then transferred to induction plates (NGM, 20 µg/mL gentamicin, 3 mg/mL L-rhamnose) until further processing.

### Fluorescent in situ hybridization

Animals were harvested with M9T, washed a couple times with PBSTw until the liquid was sufficiently clear. Paraformaldehyde was then added to final concentration of 4% v/v and incubated on a rotator at room temperature for 40 minutes. Fixed animals were then pelleted by centrifugation at 3000g for 30–60 seconds, washed 3 times with PBSTw then once with hybridization buffer (900 mM NaCl, 20 mM Tris pH 7.5, 0.01% SDS). Samples were then added with 100 mL of hybridization buffer containing appropriate probe at 10 mg/mL and incubated on thermoshaker at 46°C, 1200 rpm overnight. The next day, samples were washed 3 times, each with 1 mL of FISH wash buffer (hybridization buffer, 5 mM EDTA pH 7.5) for 30 minutes on thermoshaker at 48°C, 1200 rpm. After washing, samples were rinsed with and stored in PBSTw until imaging. To image, 5 mL of samples were mixed with 5 mL of Vectashield containing DAPI (Vector Lab) and mounted on slides.

### CFDA-tetrazine dye staining

Carboxyfluorescein diacetate succinimidyl ester (CFDA-SE, ThermoFisher C1157), methyltetrazine propylamine (mTetz, BroadPharm BP-22434), and TCO-PEG3-NHS ester (BroadPharm BP-24147) were resuspended in DMSO at concentrations of 18 mM, 20 mM, and 10 mM, respectively. Methyltetrazine propylamine was mixed with CFDA-SE in 1:1 molar ratio in amber tube and incubated at RT in dark overnight to allow for the formation of CFDA-Tetrazine.

To pre-stain animals with CFDA-Tetrazine, animals grown on OP50–1 were harvested one day before infection, washed with M9T and incubated with 15 µL of this mixture of CFDA-SE + mTetz in 3 mL M9T on a rotator at RT for 2 hr. Afterwards, animals were spun down at 400g for 1 min, rinse once with M9T, and resuspended in 1 mL of M9T and 30 µL of CFDA-SE + mTetz mixture and plated on a new NGM plate seeded with OP50–1 for further pre-staining until the next day.

On the day of infection, animals were harvested, washed 3 times with M9T and replated on a new NGM plate seeded with OP50–1. One milliliter of an overnight culture of either WT *B. atropi* or T3SS mutant was pelleted, washed 3x with M9, and incubated with TCO-PEG3-NHS at final concentration of 100 µM for 2 hr at RT on a rotator. After incubation, bacteria were washed thoroughly with an excess volume of fresh LB 3 times, 10 min each, and finally resuspended in 1 mL

of LB and overlaid on top of NGM plate with pre-stained animals. Infection was allowed to occur for 24 hr before animals were collected and proceeded with FISH.

## Transmission electron microscopy

Infected animals and non-infected control were collected with M9T, washed a couple times with PBSTw until the liquid was visibly clear, then rinsed 3 times with deionized water before fixing with room-temperature fixative solution (0.1 M sodium cacodylate buffer pH 7.5, 2.5% paraformaldehyde, 2% glutaraldehyde; prepared fresh before use) for 2 hours on a rotator at room temperature. Samples were then kept in the same fixative solution and stored at 4°C until secondary fixation and further processing by UCSD EM core facility. TEM was performed by the Electron Microscopy Facility, Department of Cellular and Molecular Medicine at UCSD.

## Dual RNA sequencing

To synchronize animals to L1 stage, animals grown on NGM plate seeded with OP50–1 were harvested when they were about to starve the plate when the population comprised mainly gravid adults and L1 larvae and embryos. Adult animals were allowed to settle at the bottom of a 15mL conical tube, and the supernatant containing only L1 larvae and embryos was collected into a different tube and spun down at 3000 g for 2 min. Pellet of L1 and eggs was washed 3 times with M9T and incubate in fresh M9T on a rotator at RT overnight to allow embryos to hatch into L1 larvae. This L1 population was then split into four equal portions and individually infected with WT *B. atropi* LUAb6 strain, filamentation-deficient strain LUAb7, and T3SS mutant strain Δ*sctC* following a standard infection procedure or plated on OP50 plate as a control. For synchronized adults, synchronized L1s were plated onto NGM plates seeded with OP50–1 and allowed to grow to day-1 adult stage, which were then harvested and performed standard infection similar to L1 larvae.

After 28 hpi, animals infected at L1 stage were harvested, washed thoroughly with M9T to get rid of extracellular bacteria outside cuticles. For animals infected at day-1 adult stage, animals were washed and further separated from eggs and larvae by several rounds of quick sedimentation by gravity. As controls, bacteria of corresponding *B. atropi* strains were grown in LB cultures overnight, and one milliliter of each culture was spun down at 4200g for 5 min, washed with PBS, and pelleted. All samples were then resuspended in TriReagent (MRC TR 118 and BP 151) and total RNAs were extracted as manufacturer's protocol.

RNA samples were ribodepleted and libraries were prepared by UCSD IGM Genomics Center using Illumina Stranded Total RNA Prep kit with Ribo-Zero plus supplemented with probes against *O. tipulae* type strain CEW1 and *B. atropi* rRNAs. Libraries were subsequently sequenced on NovaSeq X Plus platform with PE150 configuration to 200 million reads per sample.

### Bioinformatics

**Transcriptomics analyses.** Raw reads were mapped onto B. atropi LUAb4 strain complete genome (GCA_021325795.2) using hisat2 [74] using following parameters: --maxins 1000 --no-unal --dta --rna-strandness RF --mp 3,1 --sp 2,1 --rdg 4,1 --rfg 4,1 --score-min L,-7,-0.4. The resulted mapped reads were then counted for DE analysis using htseq-count [75] with parameters as follow: --max-reads-in-buffer 20000000 -a 10 -s reverse -r pos -t transcript --additional-attr=gene_name -m union. Count files were subsequently subjected to DE analysis with DEseq2 [76].

**Compositional bias analysis.** A list of amino acid sequences of hypothetical proteins of *B. atropi* was analyzed for proline-rich motif using fLPS2 [49] with the following parameters -t1e-5 -m15 -M 1000 -r P.

## Initial invasion events and normalized infected areas assays

Animals grown on NGM plate seeded with OP50–1 were harvested and infected with a pulse-infection procedure (for initial invasion events) or a standard infection (for normalized infected areas) using different T3SS putative effector knockout

strains and WT bacteria as control. Infected animals were harvested at 16 hpi or 24 hpi (initial invasion events or normalized infected areas, respectively) and processed for FISH. FISH samples were then randomly imaged with Nikon Ni-E Eclipse for 30 animals for initial invasion events or 20 animals for normalized infected areas.

For initial invasion events, number of initial invasion events were quantified as described before [10]. Briefly, only signals sufficiently away from apical surface (truly intracellular bacteria) were counted. Individual bacteria that are well separated from one another were counted as a single invasion event, whereas each of the small clusters equivalent of approximately 3–4 bacteria and well separated from either other clusters or individual bacteria was counted a one initial invasion event.

For normalized infected areas, images were analyzed with Fiji software (version 2.16.0/1.54p). Briefly, images were split into individual channel and converted into 8-bit format. Bacterial signals were adjusted for threshold using "Auto" option. Two regions of interest (ROI) were selected with segmented line tool, one outlining the body (whole body ROI) and one outlining the intestinal lumenal space and pharynx of the animals (extracellular ROI). Whole body area was measured for body ROI with "Measure" function and infected areas were measured by "Analyze particles" function within the intracellular space (XOR function of body ROI and extracellular ROI in ROI manager). Normalized infected areas are the ratios of infected areas against whole body areas.

### Anterior-posterior (A-P) normalized infection length

A-P normalized infection lengths of infected worms were measured as described before [10]. Briefly, the length of each contiguous infection in an animal was measured along the A-P axis and summed. This total infection length was normalized to the A-P length of the animal giving the normalized infection length.

### Statistics and reproducibility

All statistical analyses were performed with GraphPad Prism (version 10.2.2 (341)). For all experiments where results are presented with qualitative representative images, each experiment was repeated 2 or 3 times independently with similar results. Data from independent experiments showed comparable distributions and no systematic differences.

### Supporting information

**S1 Fig. Different phenotypes of wildtype *B. atropi* at 22 hpi by transmission electrion microscopy.** Bacteria are pseudo-colored in green. Yellow arrowheads indicate apical junctions. White arrowheads indicate basement membrane. lu, lumen, mu, muscle. Scale bars are 500 nm.
(DOCX)

**S2 Fig. Effects of putative T3SS effector knockouts.** (A) Percent animals infected when exposed to putative T3SS effector knockout strains. (B) Growth curves of different knockout strains compared to WT. Graphs show means with SD from 2 independent replicates, *, p = 0.0273, ns, non-significant by one-way ANOVA.
(DOCX)

**S3 Fig. Growth defects of overexpressing *deiA* in ΔdeiA background.** (A) ΔdeiA was complemented with a WT copy of *deiA* under control of L-rhamnose inducible promoter. Induction at different concentrations (0.5 or 3 mg/mL) results in growth defects compared to uninduced, the original knockout strain or the WT. Graph shows means with SEM from 6 technical replicates. (B) Number of initial invasion events of induced complemented ΔdeiA compared to uninduced condition and WT at 0.5 mg/mL. Graph shows means with SD from 2 independent replicates for a total of at least 13 infected animals, *, p = 0.0126, ns, non-significant by Kruskal-Wallis test followed by Dunn's tests.
(DOCX)

**S1 Table.** *B. atropi* **differentially expressed genes upregulated in vivo compared to in vitro with a more than 4 fold change and FDR<0.1%.**
(XLSX)

**S2 Table.** *B. atropi* **differentially expressed genes downregulated in vivo compared to in vitro with a more than 4 fold change and FDR<0.1%.**
(XLSX)

**S3 Table. Proline-rich motif containing T3SS effector candidates.**
(XLSX)

## Acknowledgments

This publication includes data generated at the UC San Diego IGM Genomics Center utilizing an Illumina NovaSeq X Plus.

## Author contributions

**Conceptualization:** Tuan D. Tran, Robert J. Luallen.

**Formal analysis:** Tuan D. Tran, Serena J. Meadows-Graves.

**Investigation:** Tuan D. Tran, Serena J. Meadows-Graves.

**Methodology:** Tuan D. Tran.

**Supervision:** Robert J. Luallen.

**Validation:** Amanda R. Haio, Alexander I. Varga.

**Writing – original draft:** Tuan D. Tran.

**Writing – review & editing:** Tuan D. Tran, Robert J. Luallen.

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
