## [Decision Letter · Decision Letter 0]

4 Aug 2025

PPATHOGENS-D-25-01620

A type III secretion system is required for Bordetella atropi invasion of host cells in vivo

PLOS Pathogens

Dear Dr. Luallen,

Thank you for submitting your manuscript to PLOS Pathogens. Your manuscript has been evaluated by two reviewers, whose comments are provided below. Both reviewers find the manuscript compelling and of broad interest, with data supportive of the important conclusions drawn. In particular, the mechanistic characterization of the unusual Oscheius tipple-Bordetella atropi pathogenenesis interaction provides novel insights regarding the role of the T3SS in evolutionarily diverse hosts. Please address the following issues in a revised manuscript:

1) Reviewer 1 asks about the inclusion of the full list of DEGs with fold-change and statistical significance for genes induced in B. atropi during infection, 2) Reviewer 1 asks about a rescue experiment for the deiA mutant, 3) Reviewer 1 asks for further clarification regarding EM images, and 4) Reviewer 2 has a number of suggestions for clarification and improvement in the statistical analysis and presentation of data.

Please submit your revised manuscript within 30 days Oct 03 2025 11:59PM. If you will need more time than this to complete your revisions, please reply to this message or contact the journal office at plospathogens@plos.org. Please include the following items when submitting your revised manuscript:

We look forward to receiving your revised manuscript.

Kind regards,

Dennis H Kim

Guest Editor

PLOS Pathogens

D. Scott Samuels

Section Editor

PLOS Pathogens

Sumita Bhaduri-McIntosh

Editor-in-Chief

PLOS Pathogens

orcid.org/0000-0003-2946-9497

Michael Malim

Editor-in-Chief

PLOS Pathogens

orcid.org/0000-0002-7699-2064

**Journal Requirements:**

3) We notice that your supplementary Figures are included in the manuscript file. Please remove them and upload them with the file type 'Supporting Information'. Please ensure that each Supporting Information file has a legend listed in the manuscript after the references list.

Potential Copyright Issues:

i) Figures 2A, and 3C. Please confirm whether you drew the images / clip-art within the figure panels by hand. If you did not draw the images, please provide (a) a link to the source of the images or icons and their license / terms of use; or (b) written permission from the copyright holder to publish the images or icons under our CC BY 4.0 license. Alternatively, you may replace the images with open source alternatives. See these open source resources you may use to replace images / clip-art:

2) If any authors received a salary from any of your funders, please state which authors and which funders..

**Reviewers' Comments:**

Reviewer's Responses to Questions

**Part I - Summary**

Reviewer #1: This manuscript addresses a compelling and relatively unexplored question in bacterial pathogenesis: how an intracellular nematode pathogen uses secretion systems for host cell invasion. The experiments are overall well designed, and the integration of dual transcriptomics and genetic analysis is a strength. While the findings are similar to the requirement for T3SS’s for cell invasion in other systems, the establishment of the nematode system for further research of this broad phenomena is interesting as is the identification of potential new effector proteins. Overall, I think this is a good manuscript but there are some (relatively) small additions I think would good to add to this manuscript prior to publication.

Reviewer #2: This study investigates the mechanism by which the intracellular bacterial pathogen Bordetella atropi infects its nematode host Oscheius tipulae. It builds on previous work by the same group that established this system for the study of infection processes of intracellular pathogens and revealed filamentation as a novel cell-to-cell spreading mechanisms during B. atropi infection. Here, the authors go a step further in the investigation of the bacterial factors that mediate B. atropi initial invasion of host cells. They test the involvement of B. atropi secretion systems, since these secretion systems are known to be crucial for the delivery of bacterial virulence factors to host cells. They find that the B. atropi T3SS, but not T6SS, is required for host cell invasion. A comparison of the infection process between B. atropi wildtype and T3SS mutants using both electron microscopy and confocal fluorescent microscopy revealed differences in host cell protrusions from the apical surface that engulf invading bacteria. Finally, the authors analyzed additional, putative bacterial effector genes that are upregulated in B. atropi during host infection in vivo (in comparison to in vitro) using transcriptomics. Functional analysis revealed that two of the upregulated genes, the virulence regulator BvgS (known from the genus Bordetellae and the novel deiA (decreased invasion protein A) indeed are required for cell invasion. deiA encodes a proline-rich motif containing protein, which are known to be involved in host cell invasion by intracellular bacterial pathogens.

The findings presented in this manuscript are significant and of broad interest. For many intracellular pathogens it is not well understood how exactly they hijack host cell machinery to invade host cells. This study demonstrates a system for the detailed study of the infection processes of a bacterial intracellular pathogen and its nematode host and here presents novel insights into the role of of T3SS- and T6SS-mediated pathogenesis in nematodes. It is particularly noteworthy that the authors succeeded in the challenging task of capturing the invasion process of wild-type and mutant bacteria in microscopic images. This made interesting observations possible such as the engulfment of bacteria by host cell protrusions and the direct contact of invading B. atropi cells with the host cytoplasm. The mechanism of T3SS-mediated intracellular invasion seems to be conserved and thus enable future investigations of common as well as novel principles of host-pathogen interactions in this in vivo system. The findings are thus valuable for the field and will certainly spark further studies. The publication of the current results is warranted. The manuscript is clearly written, and the experiments are conducted and presented in a convincing and comprehensive way.

**Part II – Major Issues: Key Experiments Required for Acceptance**

Reviewer #1: 1) The authors report the identification of DEGs in B. atropi in response to infection (Fig 5A-C) but there does not appear to be a supplemental table showing the DEGs or their fold change/statistical significance. Only a supplemental table of the Proline Rich Motif proteins that change. I think a full table of the DEGs should be added to the manuscript and presumably this should already be data the authors have in hand.

Reviewer #2: (No Response)

**Part III – Minor Issues: Editorial and Data Presentation Modifications**

Reviewer #1: 2) Along the same lines as point 1, it is very interesting that T3SS genes are among the most increased in abundance. Are T1SS or T2SS components also among the DEGs? Or T6SS genes? Figure 1 of this manuscript does a good job showing that T6SS’s are dispensable for this infection, but does not also KO the of T1SS or T2SS’s despite reporting that their genome analysis identified one T1SS and 1 T2SS in B. atropi. The logic for not focusing on T1SS or T2SS is that they do not inject effectors into target cells and only secrete proteins into extracellular space. However, I still think it would have been more complete to also KO a core component of the T1SS and T2SS’s given the logic presented and think the results would have been interesting either way (are/are not required). This is especially true if T1SS or T2SS genes were also among the DEGs. However, I do not feel that this is not absolutely required for the claims of this manuscript, simply that it would help with completeness and characterization given ordering of the manuscript leading with one type of SS is not required.

3) Could the authors provide a deiA rescue experiment in figure 5 analagous or similar to their rescues in Fig. 2? The observation for the deiA mutant bacteria is a core new claim and given that it’s only one mutant I think rescue or a 2nd allele should be easy enough to provide and important

4) The authors present data showing that exoY mutant bacteria produces a small by statistically significant effect on invasion. Would an deiA and exoA double mutant be even more defective in invasion? Do the authors think these two potential effectors might function in parallel?

5) I personally have trouble seeing/identifying villi confidently in some of the EM images provided in Fig 3A, particularly for the sctC mutant. If better images are available or potential better highlighting of what is being measured and how it was chosen I think it would be helpful for interpreting the quantification in Fig. 3B.

Minor point:

6) E. coli is not italicized in line 104

Reviewer #2: Figure 1 C and D, Figure 2 D, Figure 5 F and G, and Figure S2 A: Data are pooled, how was variation between independent experiments handled? (i.e was a test for batch effects done?). Please indicate in figure legends and/or materials and methods.

Figure 1 B and D What is the difference between the filamentation phenotype shown in the representative microscopy figures in Figure 1B and the A-P infection length/morphology phenotype shown in Figure 1 D? Does the data in Figure 1D represents a measurement of the length of filaments shown (representatively) in the microscopy images?

Figure 2A: I suggest to remove all the labels with abbreviations for the subunits that are not relevant since they are also not explained in the figure legend.

Figure 2B It is difficult to see that the bacterial mutants only colonize the lumen of the host intestine and are not intracellular, especially with the wildtype control missing.

Figure 2D: what is the sample size?

Figure 4 A: could the arrows also be added to the merged image? It then may be easier for the reader to see the engulfment of the bacteria by host protrusions.

PLOS authors have the option to publish the peer review history of their article (what does this mean? ). If published, this will include your full peer review and any attached files.

**Do you want your identity to be public for this peer review?** For information about this choice, including consent withdrawal, please see our Privacy Policy .

Reviewer #1: No

Reviewer #2: No

**Figure resubmission:**
---

## [Editor Report · Decision Letter 1]

29 Jan 2026

Dear Dr. Luallen,

We are pleased to inform you that your manuscript 'A type III secretion system is required for Bordetella atropi invasion of host cells in vivo' has been provisionally accepted for publication in PLOS Pathogens.

Best wishes,

Scott

D. Scott Samuels

Section Editor

PLOS Pathogens

Sumita Bhaduri-McIntosh

Editor-in-Chief

PLOS Pathogens

orcid.org/0000-0003-2946-9497

Michael Malim

Editor-in-Chief

PLOS Pathogens

orcid.org/0000-0002-7699-2064

---

## [Editor Report · Acceptance letter]

Dear Dr. Luallen,

We are delighted to inform you that your manuscript, "A type III secretion system is required for Bordetella atropi invasion of host cells in vivo," has been formally accepted for publication in PLOS Pathogens.

Best regards,

Sumita Bhaduri-McIntosh

Editor-in-Chief

PLOS Pathogens

orcid.org/0000-0003-2946-9497

Michael Malim

Editor-in-Chief

PLOS Pathogens

orcid.org/0000-0002-7699-2064